# Low-affinity integrin states have faster ligand-binding kinetics than the high-affinity state

Jing Li[1,2], Jiabin Yan[1,2], Timothy A Springer[1,2]*

[1]Program in Cellular and Molecular Medicine, Boston Children's Hospital, Boston, United States; [2]Department of Biological Chemistry and Molecular Pharmacology and Department of Pediatrics, Harvard Medical School, Boston, United States

**Abstract** Integrin conformational ensembles contain two low-affinity states, bent-closed and extended-closed, and an active, high-affinity, extended-open state. It is widely thought that integrins must be activated before they bind ligand; however, one model holds that activation follows ligand binding. As ligand-binding kinetics are not only rate limiting for cell adhesion but also have important implications for the mechanism of activation, we measure them here for integrins α4β1 and α5β1 and show that the low-affinity states bind substantially faster than the high-affinity state. On- and off-rates are similar for integrins on cell surfaces and as ectodomain fragments. Although the extended-open conformation's on-rate is ~20-fold slower, its off-rate is ~25,000-fold slower, resulting in a large affinity increase. The tighter ligand-binding pocket in the open state may slow its on-rate. Low-affinity integrin states not only bind ligand more rapidly, but are also more populous on the cell surface than high-affinity states. Thus, our results suggest that integrin binding to ligand may precede, rather than follow, activation by 'inside-out signaling.'

## Editor's evaluation

This manuscript describes a detailed measurement and calculation of integrin ligand-binding kinetics, which are very important for the understanding of integrin activation. The data clearly indicate that low-affinity-binding states with closed conformation bind ligand with the mode of 'fast on, fast off,' while the high-affinity-binding state with the open conformation shows very slow off-rate. The kinetics measurements were well designed and a lot of work was done in this study.

## Introduction

Integrins are a family of receptors that in all adherent cells in the body mechanically integrate the intracellular and extracellular environments and mediate cell migration and adhesion. Their α- and β-subunits associate noncovalently to form an extracellular ligand-binding head and multidomain 'legs' that connect to single-pass transmembrane and cytoplasmic domains with binding sites for cytoskeletal adaptor or inhibitory proteins (*Figure 1A*). Integrins populate an ensemble with three overall conformational states: the low-affinity bent-closed (BC) and extended-closed (EC) conformations and the high-affinity extended-open (EO) conformation (*Figure 1A*). The equilibrium between these conformational states is allosterically regulated by extracellular ligand binding, intracellular adaptor/inhibitor binding (*Bouvard et al., 2013*; *Iwamoto and Calderwood, 2015*), and tensile force applied by the actin cytoskeleton on the integrin β-subunit that is resisted by ligand embedded in the extracellular matrix or on cell surfaces (*Kim et al., 2011*; *Legate and Fässler, 2009*; *Li and Springer, 2017*; *Nordenfelt et al., 2016*; *Park and Goda, 2016*; *Sun et al., 2016*; *Zhu et al., 2008*; *Figure 1A*).

*For correspondence:
springer@crystal.harvard.edu

Competing interest: The authors declare that no competing interests exist.

**Figure 1.** Ligand-interaction kinetics of integrin ensembles. (**A**) Three overall integrin conformational states (*Luo et al., 2007*). Individual domains are labeled next to the extended-open (EO) state. The structural motifs that move during opening (α1-helix, α7-helix, and β6-α7 loop) are labeled in the βI domain of the extended-closed (EC) and EO state. F represents tensile force exerted across ligand–integrin–adaptor complexes by the cytoskeleton and resisted by immobilized ligand. (**B**) Reaction scheme showing the apparent 1 vs. 1 kinetics of integrin and ligand binding (left) and the scheme taking into account conformational change (right). (**C**) Fabs utilized in this study, the integrin domains they bind, and their conformational specificities.

The EO conformation has ~1000-fold higher binding affinity for ligand than the two closed conformations and is the final competent state to mediate cell adhesion and migration (*Li and Springer, 2018*; *Li et al., 2017*; *Schürpf and Springer, 2011*). Many previous studies have emphasized the importance of force in regulating integrin adhesiveness (*Alon and Dustin, 2007*; *Astrof et al., 2006*; *Li and Springer, 2017*; *Nordenfelt et al., 2016*; *Nordenfelt et al., 2017*; *Sun et al., 2019*; *Zhu et al., 2008*). Recent measurements of the intrinsic ligand-binding affinity of each conformational state and the equilibria linking them enabled a thermodynamic comparison of integrin activation models (*Li and*

*Springer, 2017*; *Li et al., 2017*). Remarkably, only the combination of adaptor binding and cytoskeletal force can activate integrins in an ultra-sensitive, switch-like manner over a narrow range of signal input that is the sine qua non of cellular signaling (*Kuriyan et al., 2012*; *Li and Springer, 2017*). The large increase in length between the bent and extended conformations (*Figure 1A*) is indispensable for switch-like integrin activation.

Despite these advances, thermodynamics cannot describe the sequence of events in a multi-step transition; furthermore, energy-driven processes such as cytoskeleton movements occur under nonequilibrium conditions. To quantitatively relate the steps involved in signal transmission across the plasma membrane in integrins, ligand-binding on- and off-rates of each conformational state are of key importance. These parameters also determine whether integrin encounter of ligand is timely and whether the ligand remains bound for a sufficiently long time for the integrin to exert its function in the presence of force. Previous representative measurements (*Dong et al., 2018*; *Kokkoli et al., 2004*; *Mould et al., 2014*; *Takagi et al., 2003*) on integrin interaction with ligand have yielded kinetics on mixtures of conformational states, that is, apparent on- and off-rates averaged over conformational states (*Figure 1B*, left). However, the ligand-binding kinetics of individual integrin conformational states (*Figure 1B*, right) remain unknown. These kinetics must be determined before we can understand how integrin function is regulated and how integrins work in concert with the cytoskeleton to provide traction for cell migration and firm adhesion for tissue integrity (*Figure 1A*).

For two classes of force-regulated adhesion molecules, each of which have a single low-affinity state and a single high-affinity state, selectins (*Phan et al., 2006*) and FimH (*Yakovenko et al., 2015*), it has been postulated that the kinetics are faster for the lower-affinity state. If after binding to the low-affinity state subsequent conformational change to the high-affinity state is rapid, fast ligand-binding kinetics to the low-affinity state efficiently couples ligand binding to stabilization by applied force of the high-affinity state, which has a long lifetime (*Yakovenko et al., 2015*). However, the ligand-binding kinetics of the states of these receptors have not yet been measured. Work from our group on integrin αVβ6 showed that removal of the hybrid domain in the αVβ6 head resulted in a 50-fold increase in affinity for ligand yet decreased the apparent on-rate of ligand binding (*Dong et al., 2018*). However, whether this is related to an increase in the population of the open conformation could not be determined due to the lack of tools to stabilize specific conformational states. There is a similar lack of measurements of state-specific ligand-binding kinetics on other type I single-pass transmembrane receptors, many of which have both inactive and active conformations. In contrast, the field is more advanced for multipass receptors, such as the β2-adrenergic receptor in the G protein-coupled receptor family, which is stabilized by binding of intracellular G proteins in a high-affinity state that has slower on- and off-rates (*DeVree et al., 2016*).

In this study, we utilized well-characterized conformation-specific Fabs (*Li and Springer, 2018*; *Li et al., 2017*; *Su et al., 2016*; *Figure 1C*) at saturating concentrations to stabilize integrins α4β1 and α5β1 into defined ensembles containing only one or two of the three integrin conformational states and measured the ligand-binding kinetics of each defined ensemble. Together with previously determined intrinsic ligand-binding affinities and populations of conformational states (*Li and Springer, 2018*; *Li et al., 2017*), our measurements enable us to define ligand-binding kinetics intrinsic to each conformational state. For each integrin, the two closed states have indistinguishable on- and off-rates for ligands. Remarkably, the on-rate for ligand of the low-affinity closed integrin conformations is ~40-fold (α4β1) or ~5-fold (α5β1) higher than that for the high-affinity EO conformation. The ~1000-fold higher affinity of the EO conformation than the closed conformations is achieved by the ~25,000-fold lower off-rate of the EO conformation for both α4β1 and α5β1 integrins. These findings suggest for two representative β1 integrins that most ligand binding occurs to the BC and/or EC states. These measurements have important implications for the order of the steps in integrin activation.

## Results
### Ligand-binding kinetics of intact α4β1 and α5β1 on cell surfaces

We measured binding kinetics of intact α4β1 on Jurkat cells to two fluorescently labeled ligands, a phenylureide derivative of Leu-Asp-Val-Pro (FITC-LDVP) and a fragment of vascular cell adhesion molecule (VCAM) containing its first two domains (Alexa488-VCAM D1D2) (*Figure 2*). Before adding ligands, cells were equilibrated with saturating concentrations of Fabs for 30 min at 22°C to stabilize

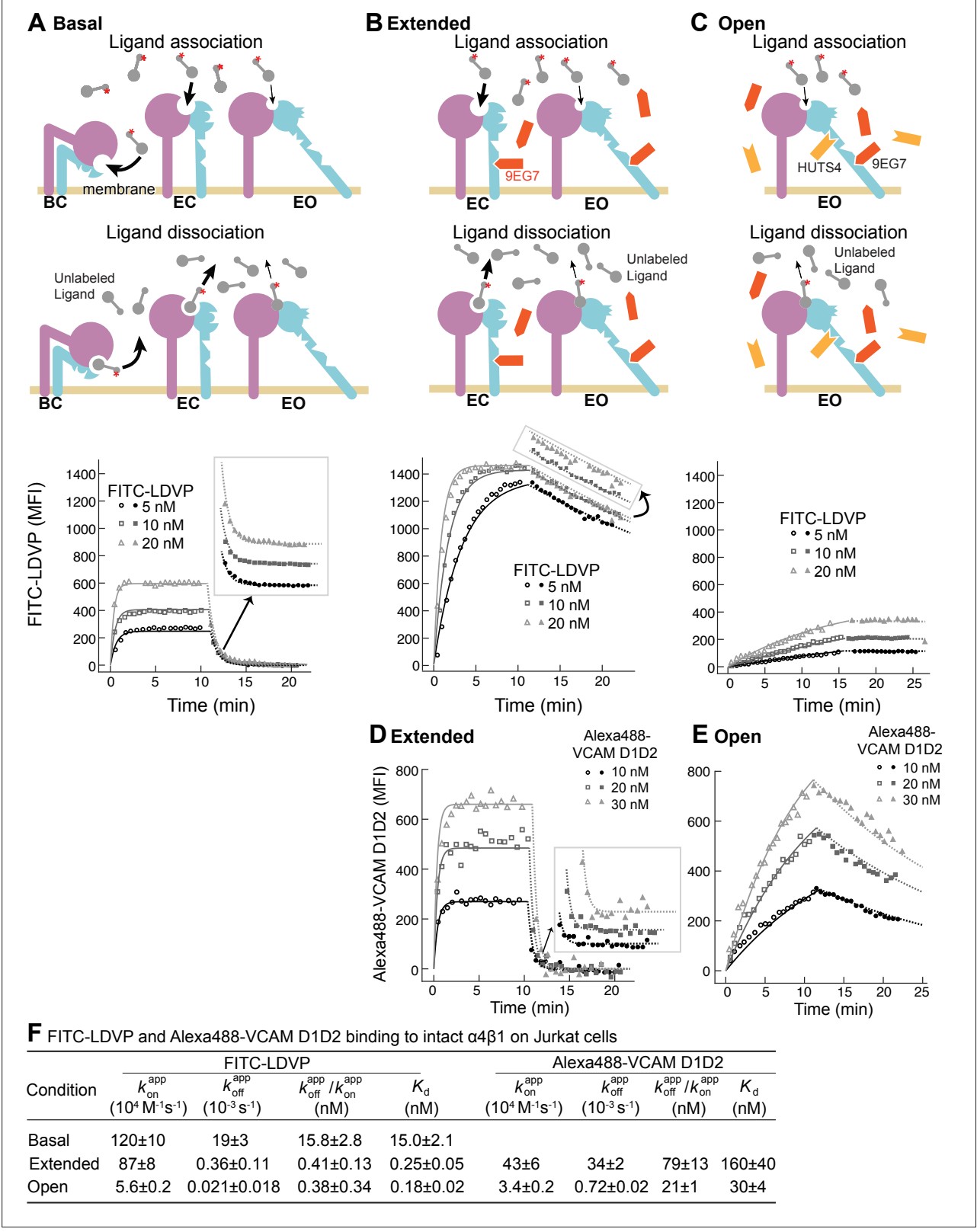

**Figure 2.** Binding kinetics of ligands to intact α4β1 on Jurkat cells. (**A–E**) Binding and dissociation of FITC-LDVP (**A–C**) and Alexa488-VCAM D1D2 (**D, E**) to α4β1 on Jurkat cells measured by flow cytometry. Cartoons in panels **A–C** show the schemes for measuring $k_{on}$ in the association phase and $k_{off}$ in the dissociation phase in the basal ensemble (**A**), the extended ensemble stabilized with Fab 9EG7 (8 μM) (**B, D**), and the open ensemble stabilized with Fabs 9EG7 (4 μM) and HUTS4 (2 μM) (**C, E**). Specific mean fluorescence intensity (MFI) with the MFI in EDTA (***Figure 2—figure supplement 1***)

*Figure 2 continued on next page*

*Figure 2 continued*

subtracted is shown as open (association) or filled (dissociation) symbols; fits are shown as thin lines. (**F**) Tabulation of results. $k_{on}^{app}$ and $k_{off}^{app}$ are from global fits of data at all ligand concentrations. $k_{off}^{app}/k_{on}^{app}$ is also shown and compared to the equilibrium $K_d$ measurements shown in Figure 4A and D of our previous work (*Li and Springer, 2018*), except for Alexa488-VCAM D1D2 binding to extended states, which was measured here (*Figure 2—figure supplement 2*). Errors for $K_d$ values are SD from three independent experiments. Kinetic constants were determined multiple times. Kinetic constants determined in a preliminary experiment were used to determine the range of concentrations and times used in the experiment shown. Each of the three concentrations in the experiment shown is an independent experiment capable of determining the reported kinetics. By globally fitting kinetic constants to all three analyte concentrations, we obtained single on- and off-rates that combine the measurements in the three independent determinations and at the same time report errors in concentrations used and model assumptions about concentration dependence. Errors for $k_{on}^{app}$ and $k_{off}^{app}$ values are standard error (SE) from global nonlinear least-square fitting of data at all ligand concentrations. From SE, the 95% confidence interval for the fitting parameter can be computed as [fitting value – 2 * SE, fitting value + 2 * SE]. Errors for $k_{off}^{app}/k_{on}^{app}$ are propagated from errors of $k_{on}^{app}$ and $k_{off}^{app}$.

The online version of this article includes the following figure supplement(s) for figure 2:

**Source data 1.** Source data for *Figure 2A*.

**Source data 2.** Source data for *Figure 2B*.

**Source data 3.** Source data for *Figure 2C*.

**Source data 4.** Source data for *Figure 2D*.

**Source data 5.** Source data for *Figure 2E*.

**Figure supplement 1.** Background binding of fluorescent ligands in the presence of 10 mM EDTA.

**Figure supplement 2.** Binding of Alexa488-VCAM D1D2 to Jurkat cells under extension-stabilizing Fabs monitored by FACS.

specific conformational states (*Li and Springer, 2018*). Integrin extension, that is, the EC and EO states, was stabilized with 8μM 9EG7 Fab, which binds to the β1-subunit knee (*Figure 2B*). The EO conformation was stabilized with a combination of 4 μM 9EG7 Fab and 2 μM HUTS4 Fab; the latter binds near the interface between the βI andhybrid domains and stabilizes the EO conformation (*Figure 2C*). Ligand-binding was monitored as mean fluorescence intensity (MFI) by flow cytometry without washing (*Figure 2*). Beginning at about 10 min, a 500-fold higher concentration of unlabeled ligand was added to measure the kinetics of dissociation. Background MFI at each fluorescent ligand concentration, measured under identical conditions except in the presence of 10 mM EDTA, showed no significant difference at different time points during the association and dissociation measurements (*Figure 2—figure supplement 1*) and was averaged across different time points and subtracted to obtain specific binding.

Under basal condition with all three integrin states present in the ensemble, binding of FITC-LDVP to Jurkat cells reached equilibrium within 3 min (*Figure 2A*). Upon addition of a 500-fold excess of LDVP, dissociation of FITC-LDVP was rapid and was 99.7% complete by 5 min (*Figure 2A*). In contrast, both binding and dissociation of FITC-LDVP were slower when only the extended conformations (EC and EO) were present on Jurkat cells (*Figure 2B*). Reaching steady state required ~5 min after addition of 20 nM FITC-LDVP, ~10 min with 10 nM ligand, and was not reached after 10 min with 5 nM ligand. After 10 min of dissociation, only 19.4% of ligand had dissociated (*Figure 2B*). Association and dissociation were even slower when only the EO conformation was present (*Figure 2C*). After 15 min of association, much less ligand had bound (*Figure 2C*) than when both EC and EO conformations were present (*Figure 2B*). Dissociation was also slower, with only 1.2% of bound ligand dissociating after 10 min (*Figure 2C*).

VCAM D1D2 binds with ~100-fold lower affinity than LDVP to α4β1 (Figure 5 in *Li and Springer, 2018*). As a result, binding to the basal ensemble was too low to measure over the noise from unbound ligand; however, we were able to measure binding kinetics to intact α4β1 stabilized in the extended (EC + EO) and EO states (*Figure 2D and E*). When the two extended conformations (EC and EO) were present, binding of all three concentrations of Alexa488-VCAM D1D2 (10 nM, 20 nM, and 30 nM) reached equilibria within 2 min. Upon addition of a large excess of LDVP, dissociation of Alexa488-VCAM D1D2 was also fast; 100% dissociated by 5 min (*Figure 2D*). Association and dissociation both became markedly slower when only the EO conformation of α4β1 was present (*Figure 2E*).

To address the generality of these results, we studied another integrin and cell type by measuring binding of a fluorescently labeled two-domain fragment of fibronectin (Alexa488-Fn3$_{9-10}$) to intact α5β1 integrin on K562 cells (*Figure 3*). The BC conformation of α5β1 integrin on K562 cells is more stable than that of α4β1 integrin on Jurkat cells (*Li and Springer, 2018*). Therefore, to assure that

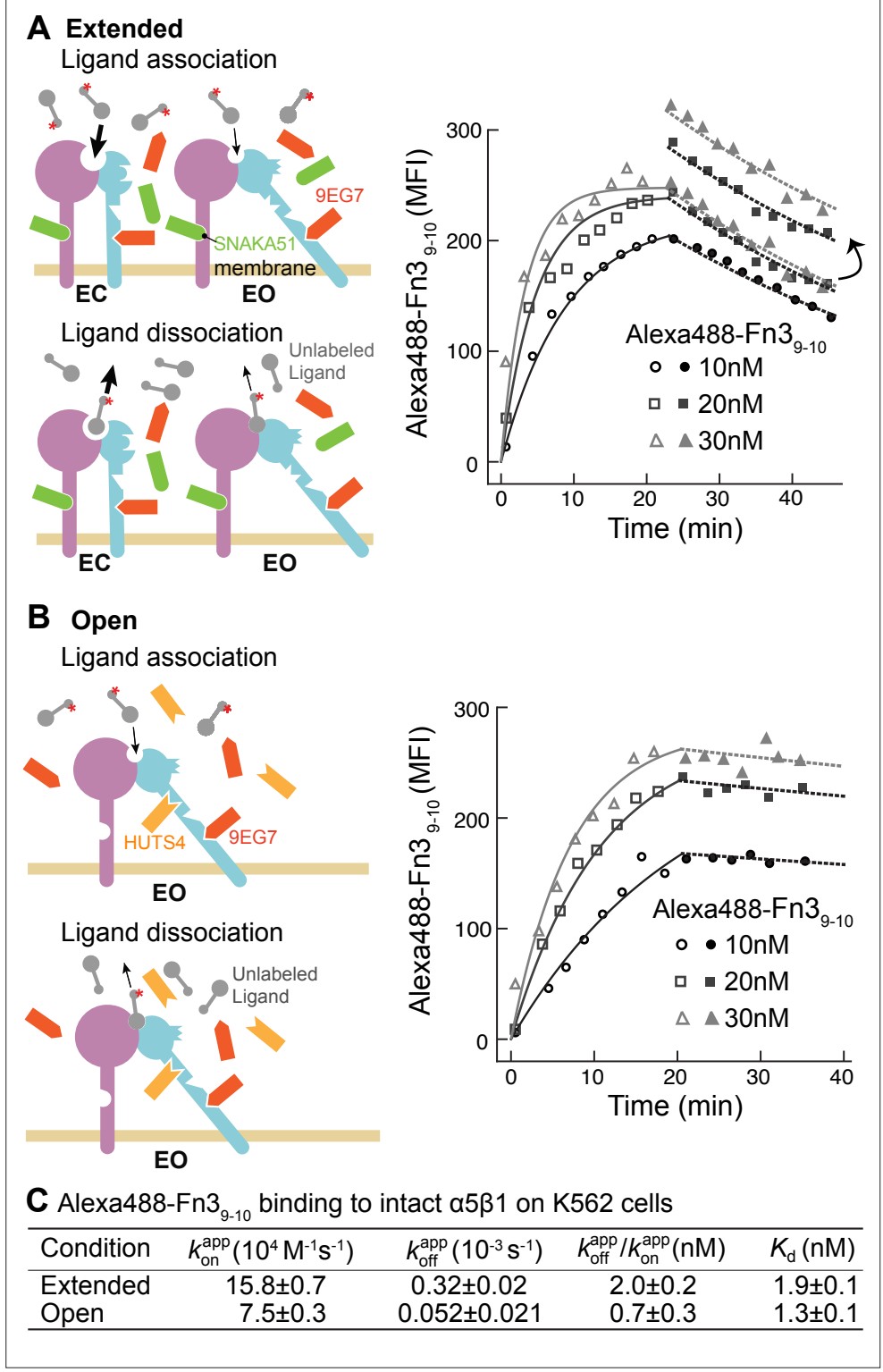

| Condition | $k_{on}^{app}$ ($10^4$ M$^{-1}$s$^{-1}$) | $k_{off}^{app}$ ($10^{-3}$ s$^{-1}$) | $k_{off}^{app}/k_{on}^{app}$ (nM) | $K_d$ (nM) |
|-----------|--------------------------|---------------------------|-----------------------------------|------------|
| Extended | 15.8±0.7 | 0.32±0.02 | 2.0±0.2 | 1.9±0.1 |
| Open | 7.5±0.3 | 0.052±0.021 | 0.7±0.3 | 1.3±0.1 |

**Figure 3.** Binding kinetics of Alexa488-Fn3$_{9-10}$ to α5β1 on K562 cells. (**A, B**) Binding of Alexa488-Fn3$_{9-10}$ to α5β1 on K562 cells measured by flow cytometry. Measurements were on integrins in extended ensembles (EC + EO ) in the presence of Fabs 9EG7 (6 μM) and SNAKA51 (2 μM) (**A**) or in the open (EO state) in the presence of Fabs 9EG7 (6 μM) and HUTS4 (2 μM) (**B**), as illustrated in the cartoons. Mean fluorescence intensity (MFI) with background in EDTA subtracted (*Figure 2—figure supplement 1*) is shown as symbols and fits are shown as lines; the association phase has open symbols and solid lines, and the dissociation phase has filled symbols and dashed lines. (**C**)

*Figure 3 continued on next page*

*Figure 3 continued*

Tabulation of results. Details about experimental repeats and kinetic parameter uncertainty, including errors and confidence intervals, are as described in *Figure 2* legend. $k_{off}^{app}/k_{on}^{app}$ is also shown with propagated error and compared to equilibrium $K_d$ measurements shown in Figure 7B of our previous study (*Li et al., 2017*).

The online version of this article includes the following figure supplement(s) for figure 3:

**Source data 1.** Source data for *Figure 3A*.

**Source data 2.** Source data for *Figure 3B*.

the extended states (EC + EO) were saturably populated, they were stabilized with a combination of two Fabs, 6 µM 9EG7 Fab and 2 µM SNAKA51 Fab (*Figure 3A*, schematic). The EO state of α5β1 was stabilized with the same combination of Fabs as used for α4β1 (*Figure 3B*, schematic). Although binding affinity was too low to measure kinetics of the basal ensemble (*Li et al., 2017*), we were able to measure Alexa488-Fn3$_{9-10}$ kinetics with the EC + EO and EO ensembles of intact α5β1 (*Figure 3*). When α5β1 was stabilized in the EO conformation, Alexa488-Fn3$_{9-10}$ bound and dissociated significantly more slowly than when both the EC and EO states of α5β1 were present in the ensemble (*Figure 3A and B*). Faster binding and dissociation of Alexa488-Fn3$_{9-10}$ from the EC + EO ensemble than EO showed that the EC state of α5β1 binds and dissociates faster than the EO state, just as found for α4β1.

To quantify the binding kinetics of intact α4β1 and α5β1 under each condition, we globally fit specific binding in both association and dissociation phases at each concentration of fluorescently labeled ligand to the 1 vs. 1 Langmuir binding model (lines in *Figure 2A-E* and *Figure 3A, B*). Apparent on- and off-rates, $k_{on}^{app}$ and $k_{off}^{app}$, were well fit, with low fitting errors relative to the measured values (*Figure 2F* and *Figure 3 C*). The ratio of the apparent off- and on-rates, $k_{off}^{app}/k_{on}^{app}$, agrees reasonably well with the previously determined equilibrium dissociation constants shown as $K_d$ in *Figure 2F* and *Figure 3C* (*Li and Springer, 2018*; *Li et al., 2017*). These agreements suggest that the 1 vs. 1 Langmuir binding model can reasonably fit the kinetic data; that is, the kinetics of conformational change between states is sufficiently fast relative to the kinetics of ligand binding to not perturb ligand-binding kinetics. Thus, the relative populations of unbound states in the ensemble were not perturbed by removal by binding to ligand.

Overall, these results show that ligand binds to and dissociates from the EO conformation more slowly than from the BC and EC conformations. The kinetics measured here for the basal and EC + EO ensembles are apparent because they include contributions from distinct conformational states present in these ensembles. In contrast, EO state kinetics are measured exactly because EO is the only state present in the EO ensemble. In the final section of 'Results,' we will use previous measurements of the populations of the states in each ensemble to calculate the on- and off-rates for conformations within mixtures of states.

## Binding kinetics of soluble α5β1 ectodomain for Fn3$_{9-10}$

We utilized biolayer interferometry (BLI) (*Wallner et al., 2013*) to measure the kinetics of binding of an ectodomain fragment of α5β1 to the biotin-labeled Fn3$_{9-10}$ fragment of fibronectin immobilized on streptavidin biosensors (*Figure 4*). The ectodomain was truncated just prior to the transmembrane domains of the α5 and β1 subunits and was expressed in a cell line containing a glycan processing mutation so that it had high-mannose rather than complex-type N-glycans. Truncation of α5β1 and high mannose glycoforms substantially raises the basal population of the EO relative to that of the EO state of intact α5β1 on cell surfaces ( Figures 4 and 7E in *Li et al., 2017*), enabling measurement here of basal ensemble Fn3$_{9-10}$ binding kinetics.

Binding kinetics were measured by transferring Fn3$_{9-10}$ biosensors to wells containing the α5β1 ectodomain in the absence or presence of conformation-stabilizing Fabs. Dissociation kinetics were measured by transfer of sensors to wells lacking the integrin but containing identical Fab concentrations (*Figure 4A–D*). Equilibrium $K_d$ values were previously shown to be independent of the Fab used to stabilize a particular state (Figure 3 in *Li et al., 2017*). However, we were concerned that binding of Fabs, particularly those that bind near ligand-binding sites, might slow kinetics and therefore tested this by varying the Fabs used to stabilize the EO state.

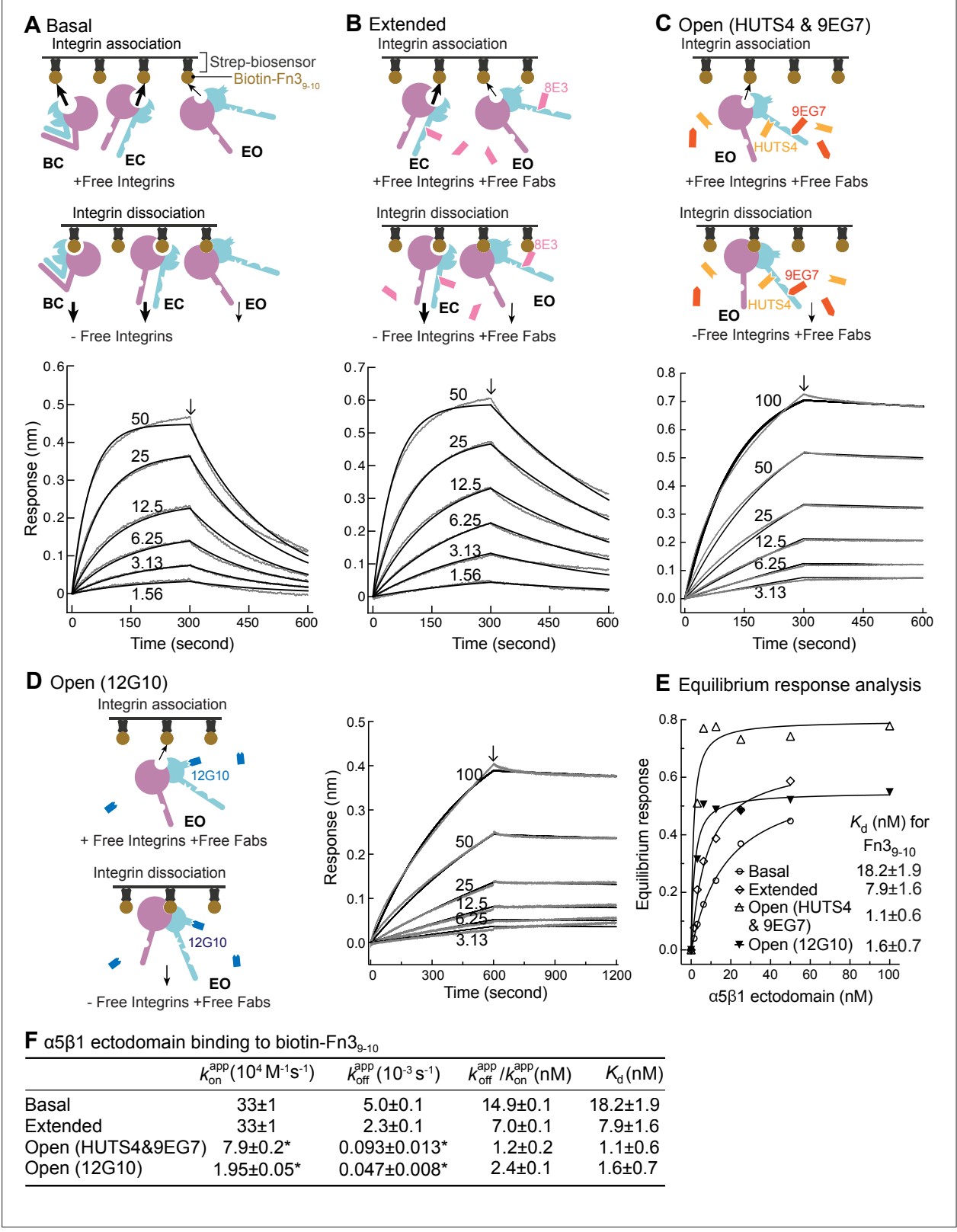

**Figure 4.** Binding kinetics of α5β1 ectodomain to Fn3$_{9-10}$. (**A–D**) Binding of unclasped high-mannose α5β1 ectodomain measured with biolayer interferometry (BLI). Schemes for measuring ligand binding and dissociation in the association phase and dissociation phase are shown in each panel. α5β1 ectodomain (analyte) at the indicated concentrations in nM was bound to biotin-Fn3$_{9-10}$ immobilized on streptavidin biosensors without Fab (**A**), with 2 μM Fab 8E3 (**B**), with 2 μM 9EG7 and 5 μM HUTS4 Fabs (**C**), or with 1 μM Fab 12G10 (**D**). Arrows mark the start of the dissociation phase.

*Figure 4 continued on next page*

*Figure 4 continued*

Response curves are in gray and fitting curves in black. (**E**) The equilibrium binding (response) was calculated from $k_{on}^{app}$ and $k_{off}^{app}$ values at each α5β1 ectodomain concentration and fit to a dose–response curve to calculate $K_d$ values. These values serve as a check on the $k_{off}^{app}/k_{on}^{app}$ values in (**F**). (**F**) $k_{on}^{app}$ and $k_{off}^{app}$ values from nonlinear least-square fit of data in panels (**A–D**) and $K_d$ values from equilibrium response analysis in panel (**E**) with 1 vs. 1 Langmuir binding model, and $k_{off}^{app}/k_{on}^{app}$ values. Errors with * are difference from the mean of two measurements at the same concentrations of α5β1 ectodomain from two independent purifications. Errors without * for $k_{off}^{app}$, $k_{off}^{app}$, and $K_d$ are SE from nonlinear least-square fits with single $k_{off}^{app}$ and $k_{off}^{app}$ as global parameters fitting to all analyte concentrations. Details about experimental repeats and uncertainties for $k_{on}^{app}$ and $k_{off}^{app}$, including errors and confidence intervals, are as described in **Figure 2** legend. Errors for $k_{off}^{app}/k_{on}^{app}$ are propagated from errors of $k_{on}^{app}$ and $k_{off}^{app}$.

The online version of this article includes the following figure supplement(s) for figure 4:

**Source data 1.** Source data for *Figure 4A*.

**Source data 2.** Source data for *Figure 4B*.

**Source data 3.** Source data for *Figure 4C*.

**Source data 4.** Source data for *Figure 4D*.

The kinetic curves showed that the α5β1 ectodomain EO state associated more slowly than the mixtures with the closed states and also dissociated more slowly (*Figure 4A-D,F*). Overall, these differences among ensembles resembled those found for the EC + EO ensemble and EO state of intact α5β1 on K562 cells and extended measurements to the basal α5β1 ensemble. The on- and off-rates of the EO state for Fn3$_{9-10}$ determined in the presence of 12G10 Fab were fourfold and twofold lower, respectively, than those determined in the presence of 9EG7&HUTS4 Fabs (*Figure 4C, D and F*). As 12G10 Fab binds close to the ligand-binding site in the β1 domain (*Figure 1A*), we use $k_{off}^{app}$ and $k_{on}^{app}$ kinetics determined with the 9EG7, 8E3, 9EG7&HUTS4 Fabs, which bind far from the ligand-binding site, for calculating true ($k_{off}$ and $k_{on}$) kinetic rates for each state in the final section of 'Results.'

## The off-rates of the closed states

Due to the low affinities of the closed states there was too little binding to directly measure $k_{on}$ or $k_{off}$ in the presence of saturating closure-stabilizing Fabs. We therefore used another approach. We first allowed ligand binding to integrins to reach steady state in the absence of a closure-stabilizing Fab. We then added different concentrations of closure-stabilizing Fab mAb13 and measured dissociation kinetics (*Figure 5* and *Figure 6*). Dissociation of the ligand from the EO state is very slow as shown above and is negligible in our experimental time scale. At high Fab mAb13 concentrations, when the EO ligand-bound state (EO•L) converts to either BC•L or EC•L (grouped together here as C•L), mAb13 Fab binds and prevents back-conversion to EO•L (*Figure 5A and B*). After saturating concentrations of Fab mAb13 are added to basal or EO+ EC ensembles pre-equilibrated with ligand, the effective off-rate is contributed by two steps, the conformational change from EO•L to C•L and the dissociation of ligand from mAb13-bound C•L (mAb13•C•L) (*Figure 5A and B*). Thus, the observed off-rate at saturating concentration of mAb13 Fab is contributed by the rates of both steps and permits the determination of the lower limit of $k_{off}^{C}$.

We measured FITC-LDVP dissociation from basal or extended ensembles of α4β1 on Jurkat cells after addition of a range of mAb13 Fab concentrations (*Figure 5A and B*). Saturable binding of mAb13 Fab to nascent cell surface C•L was evident from the approach to a plateau of $k_{off}^{max}$ (*Figure 5A–C*). The $k_{off}^{max}$ values measured for LDVP dissociation from basal and extended α4β1 ensembles on Jurkat cells were similar and within error of one another, with an average of ~120 * 10$^{-3}$ /s (*Figure 5C*).

Similarly, we measured Fn3$_{9-10}$ dissociation from basal or extended ensembles of the α$_5$β$_1$ ectodomain (*Figure 6*). The effect of mAb13 Fab on increasing $k_{off}$ was saturable, as shown by approach to a plateau (*Figure 6A–C*). The fit to a saturation dose-response curve yielded $k_{off}^{max}$ values for the basal and extended ensembles of (1600 ± 100) * 10$^{-3}$ /s and (1900 ± 100) * 10$^{-3}$/s, respectively (*Figure 6C*).

## Calculation of ligand-binding kinetics from ensemble measurements

We directly measured the ligand-binding and dissociation kinetics for the EO state of α4β1 and α5β1 (*Figure 2C,E*, *Figure 3B*, *Figure 4C*). In contrast, kinetics for the BC and EC states were only measured within ensembles. Their kinetics are convoluted in two respects. First, measurements on ensembles contain kinetics contributed by all states within the ensemble. Second, apparent association and dissociation kinetics may each contain a contribution from the kinetics of conformational

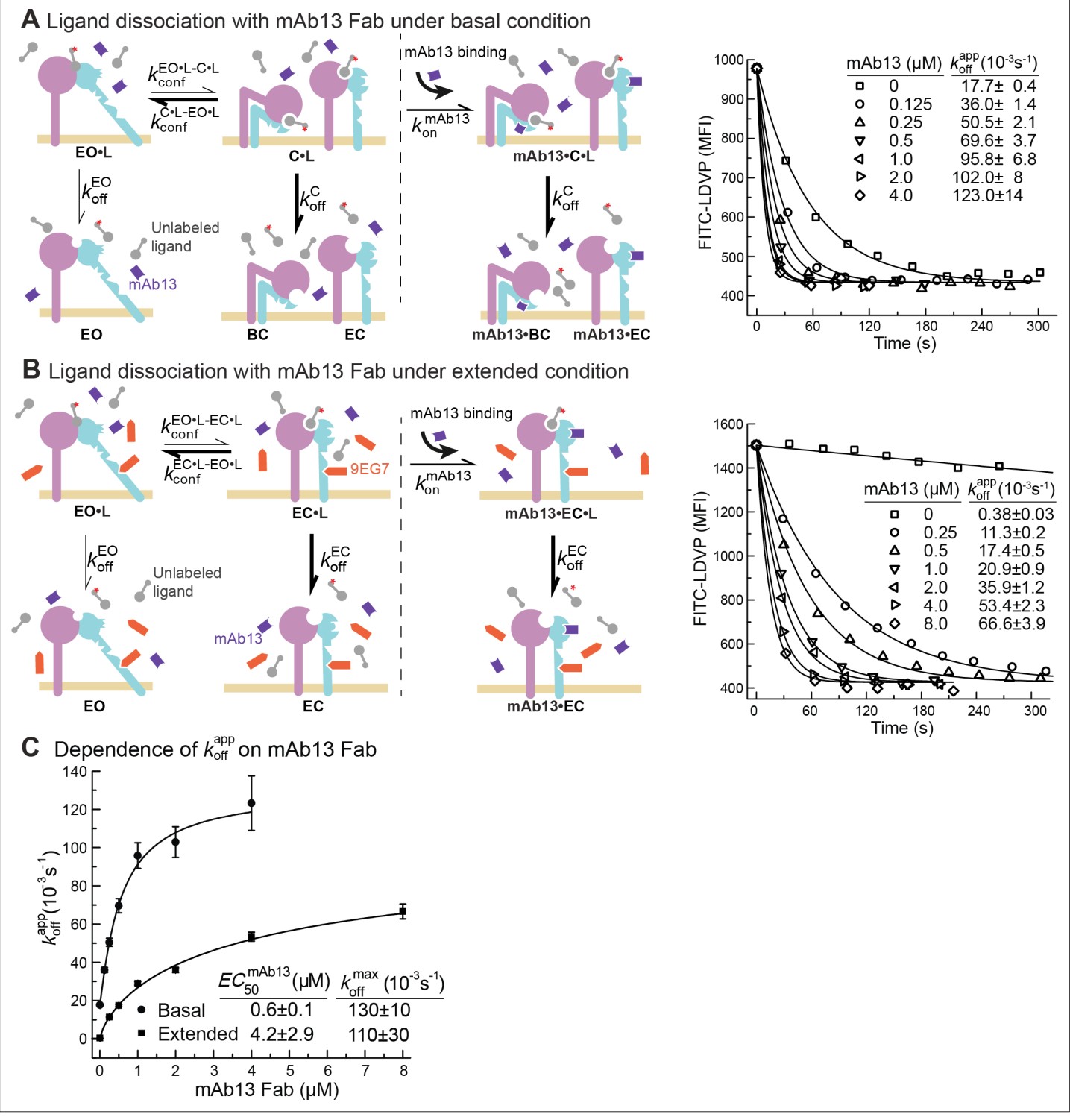

**Figure 5.** Dissociation of FITC-LDVP from α4β1 on Jurkat cells in the presence of closure-stabilizing Fab. (**A, B**) FITC-LDVP dissociation from basal or extended ensembles of intact α4β1 on Jurkat cells measured using flow cytometry. FITC-LDVP (20 nM) was incubated with Jurkat cells in the absence (**A**) or in the presence of extension-stabilizing Fab 9EG7 (4 μM) (**B**) for 10 min to reach steady state. Then, 10 μM unlabeled LDVP, together with indicated concentrations of mAb13 Fab, were added. Observed mean fluorescence intensity ($MFI_{obs}$) values as a function of time at indicated mAb13 Fab concentrations were globally fitted to $MFI_{obs} = MFI_0 * e^{-k_{off}^{app} \cdot t} + MFI_{background}$, with $MFI$ at the start of dissociation ($MFI_0$) and background $MFI$ ($MFI_{background}$) as shared parameters and $k_{off}^{app}$ as the individual fitting parameter at each mAb13 Fab concentration. (**C**) Dependence of $k_{off}^{app}$ on mAb13 Fab concentration. $k_{off}^{app}$ at each mAb13 Fab concentration in panels (**A**) and (**B**) were fitted to dose–response curves to determine the maximum off-rate at saturating mAb13 Fab concentration, $k_{off}^{max}$, and the mAb13 Fab concentration when the off-rate reaches half of the maximum, $EC_{50}^{mAb13}$. Experiments in panels (**A**) and (**B**)

*Figure 5 continued on next page*

*Figure 5 continued*

were carried out twice, first with 0, 1, 2, and 4 μM mAb13 Fab for 120 s, and then with the conditions shown, with similar $k_{\text{off}}^{\text{app}}$ values in each experiment. All errors are SE from nonlinear least-square fits.

The online version of this article includes the following figure supplement(s) for figure 5:

**Source data 1.** Source data for *Figure 5A*.

**Source data 2.** Source data for *Figure 5B*.

change (*Figure 1B*). *Figure 1B* (left) shows apparent on- and off-rates, and *Figure 1B* (right) shows all the actual pathways by which ligand binding and dissociation can occur, which include all known integrin conformational states and the kinetics of conformational change between them. Furthermore, after ligand binding to the closed states, rapid conformational change to the EO state occurs and is responsible for our ability to measure the kinetics of binding as a result of accumulation of ligand-bound integrin in the EO state.

The underlying assumption for deconvoluting the kinetics of the closed states is that if integrin conformational transition kinetics (BC ⇌ EO, EC ⇌ EO, BC ⇌ EC, BC•L ⇌ EO•L, EC•L ⇌ EO•L, BC•L ⇌ EC•L) are sufficiently fast so that the populations of the three integrin states do not deviate significantly during our experiments from the equilibrium values of the populations, then measured kinetics will not be significantly limited by conformational transition kinetics. In this case, both free integrins and ligand-bound integrins can be considered as readily equilibrated among their conformational states, and ligand binding coupled with integrin conformational changes can be approximated by the apparent 1 vs. 1 reaction between integrin and ligand (this allows the double tildes in Eqs. 1–4 in *Figure 7A* to be treated as equal signs). All on- and off-rates measured here were well fit with the 1 vs. 1 Langmuir binding model (*Figure 2A-E*, *Figure 3A,B*, *Figure 4A,B,C,D*, *Figure 5A, B*, *Figure 6A, B*), supporting this assumption. Moreover, reasonable agreement between the ratios of the apparent off- and on-rates, $k_{\text{off}}^{\text{app}}/k_{\text{on}}^{\text{app}}$, and previously determined equilibrium dissociation constants, $K_{\text{d}}$, (*Figure 2F*, *Figure 3C*, *Figure 4F*), validates the assumption that the apparent on- and off-rates ($k_{\text{on}}^{\text{app}}$ and $k_{\text{off}}^{\text{app}}$) for each defined ensemble can be approximated by the on- and off-rates of each state weighted by its population in the ensemble (*Figure 7A*, Eqs. 1–4). The population of the integrin states in the absence of ligand (BC, EC, and EO) and in the presence of saturating concentrations of ligand (BC•L, EC•L, and EO•L) was calculated based on the previously determined population and ligand-binding affinity of each state (*Figure 7—figure supplement 1B*, Eqs. S5–S10) in the respective integrin α4β1 and α5β1 preparations (*Li and Springer, 2018*; *Li et al., 2017*) and are shown in *Figure 7B*.

On- and off-rates for each α4β1 and α5β1 integrin state on intact cells and for the purified α5β1 ectodomain are summarized in *Figure 7C*. Values are best determined, that is, with the lowest errors, for the directly determined EO state on- and off-rates. Errors were higher for the BC and EC states, particularly for $k_{\text{off}}$. Therefore, $k_{\text{off}}$ values for each state were also calculated from $k_{\text{off}} = K_{\text{d}} * k_{\text{on}}$ (*Figure 7C*), where $K_{\text{d}}$ is from equilibrium measurements (*Li and Springer, 2018*; *Li et al., 2017*). The $k_{\text{off}}$ values of each state determined from these two strategies agree well with one another for each integrin-ligand pair. We do not report in *Figure 7C* the lower limit of $k_{\text{off}}^{\text{C}}$ and $k_{\text{off}}^{\text{EC}}$ approached by measuring dissociation in the presence of a closure-stabilizing Fab in *Figure 5* and *Figure 6*; however, these values were comparable to those calculated from measurements on ensembles as described in this section or calculated from $k_{\text{off}} = K_{\text{d}} * k_{\text{on}}$.

## Discussion

### Strengths and limitations of different approaches to studying integrin conformational states

Employing conformation-specific Fabs against the integrin β1 subunit to stabilize integrins into defined ensembles, we determined the on- and off-rates of each integrin conformational state. The conformational specificities of these Fabs were determined with negative stain EM (nsEM) with the integrin α5β1 ectodomain (*Su et al., 2016*). All antibodies used here were 100% conformation-specific; this conformation was present in all EM class averages of identifiable integrin conformation with bound Fab. Specificity was further verified by ligand-binding affinity measurements in the presence of saturating concentrations of Fab (*Li et al., 2017*). In the latter study, we used between 2 and

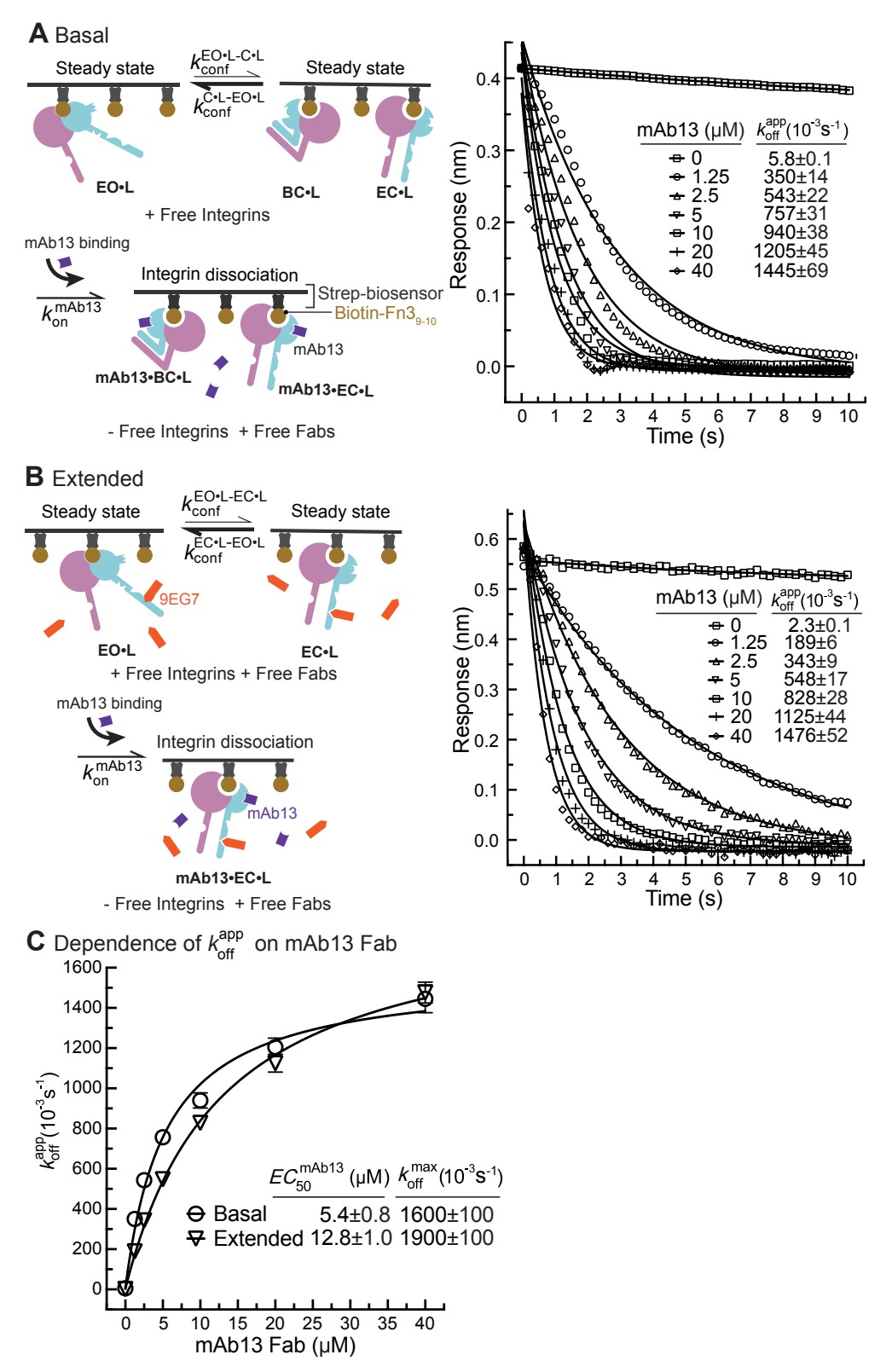

**Figure 6.** Dissociation of α5β1 ectodomain from biotin-Fn3$_{9-10}$ in the presence of closure-stabilizing Fab. (**A, B**) Unclasped high-mannose α5β1 ectodomain dissociation from biotin-Fn3$_{9-10}$ immobilized on streptavidin biosensors was monitored by biolayer interferometry (BLI). Reaction schemes are illustrated in each panel. 50 nM α5β1 ectodomain was incubated with biotin-Fn3$_{9-10}$ biosensors for 10 min to reach steady-state binding in the absence

*Figure 6 continued on next page*

*Figure 6 continued*

(**A**) or the presence of 2 μM 9EG7 Fab (**B**). Biosensors were then transferred into wells lacking the α5β1 ectodomain in the presence or absence of 9EG7 Fab as before and also containing the indicated concentrations of mAb13 Fab for measurement of dissociation. The observed response ($R_{obs}$) at each mAb13 Fab concentration as a function of time was individually fitted to the single exponential, $R_{obs} = R_0 * e^{-k_{off}^{app} \cdot t}$, for the initial response at the start of dissociation ($R_0$) and $k_{off}^{app}$. (**C**) Determination of $k_{off}^{max}$ at saturating mAb13 Fab concentration. $k_{off}^{app}$ was fit to mAb13 Fab concentration using a dose–response curve for the maximum off-rate at saturating mAb13 Fab concentration to determine $k_{off}^{max}$. The mAb13 Fab concentration when the off-rate reaches half of the maximum, $EC_{50}^{mAb13}$, was also determined. Experiments in panels (**A**) and (**B**) were carried out twice, first with 0, 2, 5, and 10 mAb13 Fab for 120 s, and then with the conditions shown, with similar $k_{off}^{app}$ values in each experiment. Errors are SE from nonlinear least-square fits.

The online version of this article includes the following figure supplement(s) for figure 6:

**Source data 1.** Source data for *Figure 6A*.

**Source data 2.** Source data for *Figure 6B*.

4 independent antibodies or antibody combinations, often binding to distinct domains, to stabilize α5β1 ectodomain ensembles containing the EO, EC, EO + EC, and EC + BC states. In all cases, independent antibodies that stabilized the same conformational state(s) as determined by EM yielded similar ensemble $K_d$ values (Figure 3 in *Li and Springer, 2018*; *Li et al., 2017*), validating stabilization of the same state(s), and supporting the assumption that these states resembled those in the absence of Fab. Quantitatively, the antibodies must be highly state-specific in order to give large shifts in affinities, to give consistent intrinsic affinities on constructs with large differences in basal affinities, and to give similar intrinsic affinities using Fabs to distinct epitopes. The conformational specificity of these antibodies and their use at sufficient concentration to saturate these states are important for our previous thermodynamic studies ( *Li and Springer, 2018*; *Li et al., 2017*) and the current kinetic study.

It is common in cryoEM and nsEM to form complexes in a large excess of one or more components, and then separate the complex from free components by gel filtration. During gel filtration and up until cryo-cooling or fixation by negative stain, complexes will dissociate because the law of mass action requires co-existing free components to stabilize complex formation, as defined by [I•L] = ([I].[L])/$K_d$ where I and L are integrin and ligand or Fab, respectively. Stabilizing integrins ≥ 99% in the desired conformations in this study required the use of Fabs at concentrations 80-fold higher than their EC50 values for each integrin preparation (Table S1 in *Li and Springer, 2018*; *Li et al., 2017*). For stabilizing intact α4β1 and α5β1 in extended and open states, some Fab EC50 values were in the hundreds of nM range, which required a combination of two compatible Fabs to conserve protein supplies. Compared to the integrin that was being stabilized here, Fabs were maintained throughout our experiments at molar excesses ranging from 9- to 48,000-fold over the integrin. This contrasts to the 1:1 ratio in integrin complexes isolated from free components for ns and cryoEM. Such complexes are out of equilibrium and must dissociate, the rate and extent of which are governed by the kinetics and equilibria such as measured here and previously, respectively (*Li and Springer, 2018*; *Li et al., 2017*).

To prevent confusion, we should discuss a recent elegant cryoEM structure of intact integrin α5β1 in nanodiscs bound both to TS2/16 Fv-clasp and Fn3$_{7-10}$ that showed for the first time that α5β1 recognizes both the RGD motif in Fn3 domain 10 and the synergy site in Fn3 domain 9 of fibronectin (*Schumacher et al., 2021*). Conformational equilibria of the α5β1 ectodomain determined with Fn3$_{9-10}$ and RGD peptides as ligands were similar (Figure 4 in *Li et al., 2017*), which makes it important to note that the claim that RGD, because it lacks the synergy site, cannot stabilize α5β1 headpiece opening, was not supported by cited articles (*Schumacher et al., 2021*). Addition of 1 mM RGD peptide to the α5β1 headpiece resulted in the presence of the headpiece in both the open and closed conformations as shown by nsEM (Figure 2B in *Takagi et al., 2003*). At saturating concentrations of ligands, integrin preparations vary in the percentage of ligand that is bound to each state (*Figure 7B*; see *Figure 7—figure supplement 1* for a wide range of integrin preparations). The α5β1 headpiece used in *Takagi et al., 2003* is ~50% open when saturably bound to a cyclic RGD peptide, in agreement with the nsEM results. The failure to open the closed α5β1 headpiece in crystals soaked with RGD was to be expected because the lattice contacts frustrated opening and in one publication a closed state-specific Fab was also bound (*Nagae et al., 2012*; *Xia and Springer, 2014*). In recent unpublished work, we found that while Mn²⁺ increases population of the EO state of cell surface

**A** Strategy for determining intrinsic on- and off rates of each integrin state

| Condition | States present in the ensemble | $k_{on}^{app}$ and $k_{off}^{app}$ for ligand |
|---|---|---|
| Open | EO<br>EO·L | $k_{on}^{EO}$<br>$k_{off}^{EO}$ |
| Extended | EC, EO<br>EC·L, EO·L | $k_{on}^{app(EC+EO)} \approx \dfrac{P^{EC}}{P^{EC}+P^{EO}}k_{on}^{EC} + \dfrac{P^{EO}}{P^{EC}+P^{EO}}k_{on}^{EO}$ (Eq.1)<br>$k_{off}^{app(EC+EO)} \approx \dfrac{P^{EC\cdot L}}{P^{EC\cdot L}+P^{EO\cdot L}}k_{off}^{EC} + \dfrac{P^{EO\cdot L}}{P^{EC\cdot L}+P^{EO\cdot L}}k_{off}^{EO}$ (Eq.2) |
| Basal | BC, EC, EO<br>BC·L, EC·L, EO·L | $k_{on}^{app(BC+EC+EO)} \approx P^{BC}k_{on}^{BC} + P^{EC}k_{on}^{EC} + P^{EO}k_{on}^{EO}$ (Eq.3)<br>$k_{off}^{app(BC+EC+EO)} \approx P^{BC\cdot L}k_{off}^{BC} + P^{EC\cdot L}k_{off}^{EC} + P^{EO\cdot L}k_{off}^{EO}$ (Eq.4) |

**B** Conformational state populations (%) in absence of ligand and when bound to ligand at saturating concentrations

| Ligand bound | Ensemble | intact α4β1-Jurkat | | | intact α5β1-K562 | | | α5β1-ectodomain | | |
|---|---|---|---|---|---|---|---|---|---|---|
| | | BC | EC | EO | BC | EC | EO | BC | EC | EO |
| - | Basal | 98.5 | 0.4 | 1.1 | 99.84 | 0.05 | 0.11 | 64.3 | 31.1 | 4.6 |
| - | Extended | 0 | 28 | 72 | 0 | 30 | 70 | 0 | 87.1 | 12.9 |
| - | Open | 0 | 0 | 100 | 0 | 0 | 100 | 0 | 0 | 100 |
| + | Basal | 10.9 | 0.04 | 89.0 | 22.6 | 0.01 | 77.4 | 0.5 | 0.2 | 99.3 |
| + | Extended | 0 | 0.05 | 99.95 | 0 | 0.01 | 99.99 | 0 | 0.2 | 99.8 |
| + | Open | 0 | 0 | 100 | 0 | 0 | 100 | 0 | 0 | 100 |

**C** Intrinsic on- and off- rates of each integrin state for ligand

| | | BC | EC | EO |
|---|---|---|---|---|
| Intact α4β1 for LDVP | $k_{on}$ $(10^4 M^{-1}s^{-1})^a$ | 120±20 | 300±170 | 5.6±0.2 |
| | $k_{off}$ $(10^{-3}s^{-1})^a$ | 170±60 | 640±590 | 0.021±0.018 |
| | $K_d$ $(nM)^b$ | 130±40 | 130±40 | 0.18±0.02 |
| | $k_{off}=K_d{}^*k_{on}(10^{-3}s^{-1})^c$ | 160±60 | 390±250 | 0.010±0.001 |
| Intact α4β1 for VCAM D1D2 | $k_{on}$ $(10^4 M^{-1}s^{-1})^a$ | | 150±90 | 3.4±0.2 |
| | $k_{off}$ $(10^{-3}s^{-1})^a$ | | 57000±23000 | 0.72±0.02 |
| | $K_d$ $(nM)^b$ | | 22000±7000 | 30±4 |
| | $k_{off}=K_d{}^*k_{on}(10^{-3}s^{-1})^c$ | | 33000±22000 | 1.0±0.1 |
| Intact α5β1 for Fn3$_{9-10}$ | $k_{on}$ $(10^4 M^{-1}s^{-1})^a$ | | 35±6 | 7.5±0.3 |
| | $k_{off}$ $(10^{-3}s^{-1})^a$ | | 1900±1200 | 0.052±0.021 |
| | $K_d$ $(nM)^b$ | | 4000±2100 | 1.3±0.1 |
| | $k_{off}=K_d{}^*k_{on}(10^{-3}s^{-1})^c$ | | 1400±800 | 0.098±0.001 |
| α5β1 ecto for Fn3$_{9-10}$ | $k_{on}$ $(10^4 M^{-1}s^{-1})^a$ | 34±2 | 37±2 | 7.9±0.2 |
| | $k_{off}$ $(10^{-3}s^{-1})^a$ | 740±400 | 1000±600 | 0.093±0.013 |
| | $K_d$ $(nM)^b$ | 3400±2600 | 3400±2600 | 1.1±0.6 |
| | $k_{off}=K_d{}^*k_{on}(10^{-3}s^{-1})^c$ | 1200±900 | 1300±1000 | 0.085±0.046 |

**D** Asp-binding pocket is tighter in the open state

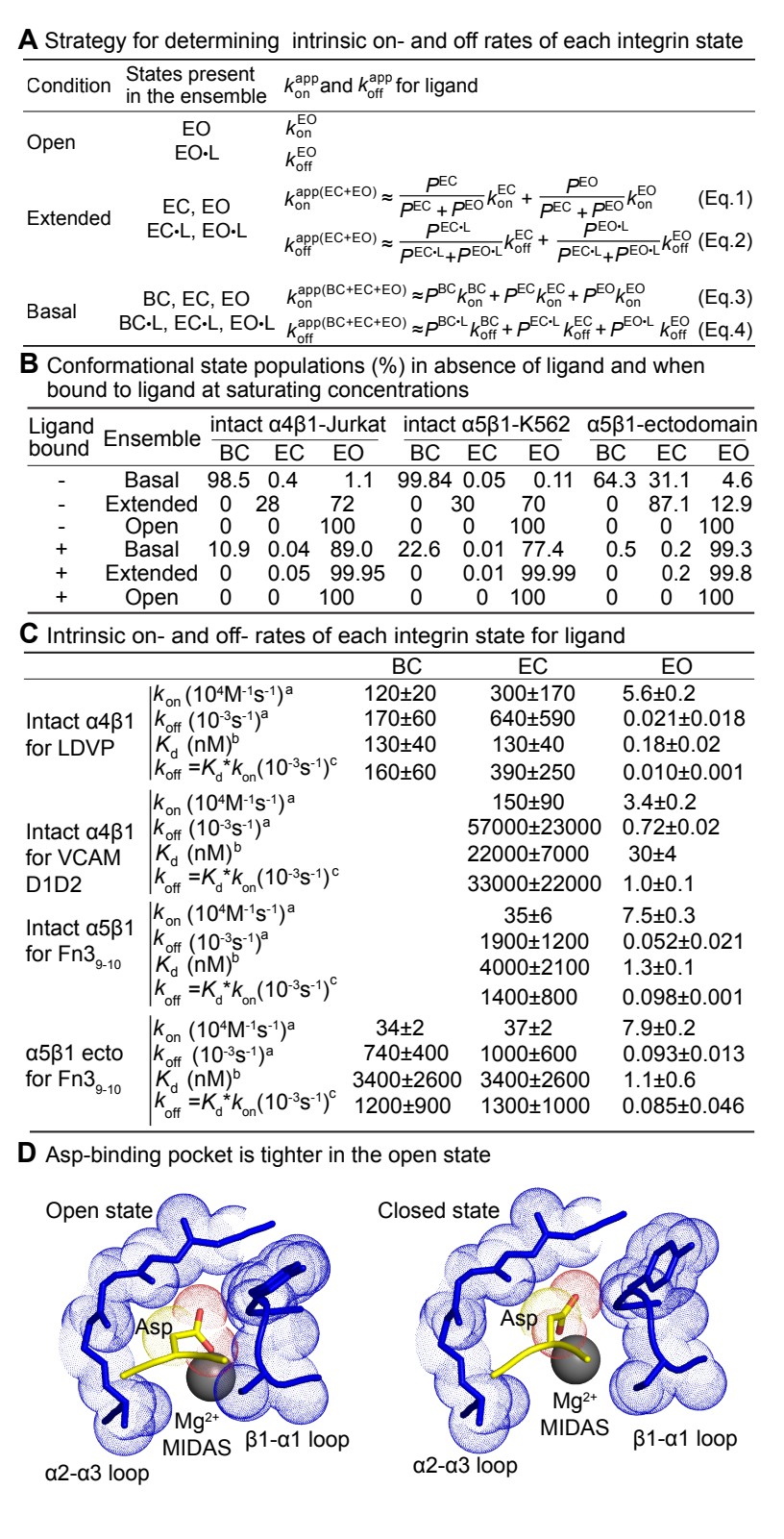

Open state · Asp · Mg²⁺ MIDAS · β1-α1 loop · α2-α3 loop

Closed state · Asp · Mg²⁺ MIDAS · β1-α1 loop · α2-α3 loop

**Figure 7.** Ligand-binding kinetics of each integrin state. (**A**) Integrin α4β1 and α5β1 ensembles utilized in this study to measure ligand-interaction kinetics and equations to relate the apparent on- and off-rates with the on- and off-rates for each conformational state. (**B**) Conformational state populations in the absence of ligand and when saturably bound to ligand. Previously reported populations for integrins in the absence of ligand and their

*Figure 7 continued on next page*

*Figure 7 continued*

affinities for ligand were used with Eqs. S5–S10 in **Figure 7—figure supplement 1B** to calculate the populations in saturating ligand of ligand-bound integrin states in each type of ensemble studied here. The population of intact α4β1 on Jurkat cell surface in the absence of ligand is reported in Figure 4D of **Li and Springer, 2018**; population of intact α5β1 on K562 cell surface and α5β1 ectodomain in the absence of ligand is reported in Figure 7 and Figure 4, respectively, in **Li et al., 2017**. The fold-difference in ligand-binding affinity for the open and closed states that was used to calculate the populations for the ligand-saturated states is described in **Figure 7—figure supplement 1**. (**C**) Values of $k_{on}$ and $k_{off}$ for conformational states of four integrin-ligand pairs. As discussed in the text and 'Materials and methods,' kinetic measurements on the extended-open (EO) state and the extended and basal ensembles were used with equations in panel (**A**) to calculate kinetics of the bent-closed (BC) and extended-closed (EC) states. The errors for directly measured values were fitting errors from nonlinear least-square fit; the errors for calculated BC and EC values were propagated. [a]Intrinsic rates of EO state were from measurements in the presence of HUTS4 and 9EG7 Fabs in **Figures 2–4**, and intrinsic rates for BC and EC states were calculated with Eqs. 1–4 in panel (**A**). [b]From previous equilibrium measurements described in the legends for **Figures 2 and 3**. [c]Calculated as shown from the product of equilibrium $K_d$ and $k_{on}$. (**D**) Comparison of Asp-binding pocket in the open state (PDB: 3ze2 chains C + D) and closed state (PDB: 3zdy chains C + D) of integrin αIIBβ3 (**Zhu et al., 2013**). The pocket in the β3 βI domain is shown with backbone and nearby sidechains in blue stick and blue dot surfaces and the MIDAS $Mg^{2+}$ ion as a silver sphere. The ligand backbone loop is shown in yellow and its sidechain in stick with yellow carbons and red carboxyl oxygens. The Asp sidechain Cβ carbon and carboxyl oxygens are shown as yellow and red dot surfaces, respectively.

The online version of this article includes the following figure supplement(s) for figure 7:

**Figure supplement 1.** Ligand-interaction kinetics of integrin ensembles.

---

α5β1, the large majority remains in the BC state (Anderson, Li, and Springer), in excellent agreement with the predominance of the BC state in $Mn^{2+}$ in α5β1 in nanodiscs (Fig. 6 in **Schumacher et al., 2021**). Additionally, while α5β1 is less bent than some other integrins (**Schumacher et al., 2021**; **Su et al., 2016**), it still has distinct bent and extended conformations and it thus remains appropriate to describe α5β1 as having a BC state.

## Intrinsic ligand-binding kinetics of integrin conformational states

The advance in this study is the measurement of ligand binding and dissociation kinetics of integrin conformational states for the first time. Furthermore, we have made the surprising observation that the low-affinity integrin BC and EC states have faster on-rates than the high-affinity EO state. While we compare $k_{off}/k_{on}$ to previously determined equilibrium $K_d$ values, that is only to demonstrate that the values we have determined are not affected by slow equilibration among the conformational states, which would have invalidated our assumption that the apparent ligand-binding kinetics are the composites of the ligand-binding kinetics of each state at equilibrium.

Previous studies on α4β1 and α5β1 integrins showed that their affinities were intrinsic to the type of integrin, the ligand, and the state; that is, independent of the type of integrin preparation: cell surface, soluble ectodomain or headpiece, or N-glycosylation status (**Li and Springer, 2018**; **Li et al., 2017**). Similar to intrinsic affinities, the results here on ligand-binding kinetics were consistent with on-rates and off-rates that are intrinsic to integrin conformational states. On-rates for Fn3$_{9-10}$ binding to EO states of intact α5β1 on cell surfaces and the α5β1 ectodomain were identical and off-rates differed by only 1.8-fold. Similarly, on- and off-rates for the EC state of the intact cell-surface and ectodomain forms of α5β1 differed only by 1.1-fold and 1.9-fold, respectively. Moreover, BC and EC states for both types of integrins had similar kinetics, showing that ligand-binding on- and off-rates were determined by whether the headpiece was closed or open, and not by extension. These findings are in agreement with the essentially identical intrinsic affinities of the two integrin α5β1 closed states (**Li and Springer, 2018**; **Li et al., 2017**). In further agreement, crystal structures of the integrin αIIbβ3 ectodomain in the BC state and of the αIIbβ3 closed headpiece fragment, which has no interactions with the lower legs and thus serves as a model for the EC conformation (**Zhu et al., 2008**; **Zhu et al., 2013**), show essentially identical conformations of the ligand-binding site.

We checked whether kinetics might be influenced by bound Fabs, in contrast to equilibria, which showed no dependence (**Li et al., 2017**). Among two Fabs used to stabilize the EO state, ligand association and dissociation was slower with 12G10, which binds near the ligand-binding site in the βI domain than with HUTS4, which binds distally in the hybrid domain (**Figure 4F**). As Fabs generally

decrease dynamic protein motions in their epitopes (*Wei et al., 2014*) and may also sterically slow binding, the kinetics measured using HUTS4 Fab more likely approximate integrin kinetics in the absence of Fab and are reported in *Figure 7C*.

The kinetics of the EC and BC states were calculated from measurements on extended or basal ensembles after correction for the kinetics in these ensembles contributed by the EO state. As a check on these measurements, we also measured $k_{off}$ in the presence of mAb13 Fab, which after conformational conversion of EO•L to EC•L + BC•L trapped the closed states so that their dissociation could be measured. The lower limit of $k_{off}^C$ and $k_{off}^{EC}$ determined from these experiments (*Figures 5C and 6C*) is in good agreement with off-rates of the closed states calculated from ensembles and also from $K_d • k_{on}$ (*Figure 7C*).

Typical protein-protein on-rates as found for antibody-antigen interactions are in the range of $10^5$–$10^6$ $M^{-1}$ $s^{-1}$ (*Alsallaq and Zhou, 2008*). The on-rates for the BC and EC states were in this range. In contrast, the on-rates for EO states for the corresponding integrin-ligand pairs were slower and suggest a hindrance to ligand binding. The open conformation of integrins has a tighter ligand-binding pocket around the metal ion-dependent adhesion site (MIDAS) than the closed state (*Dong et al., 2017*; *Nagae et al., 2012*; *Schumacher et al., 2021*; *Xia and Springer, 2014*; *Xiao et al., 2004*; *Zhu et al., 2013*; *Figure 7D*). Movement of the β1-α1 loop toward the ligand and the MIDAS $Mg^{2+}$ ion upon βI domain opening partially buries the $Mg^{2+}$ ion and is expected to slow binding of the Asp sidechain, which must fit into a tight pocket with a specific geometry dictated by partially covalent and highly directional Asp sidechain metal coordination and hydrogen bonds to the β1-α1 loop backbone amide nitrogens.

The ~1000-fold higher affinities of the EO than the closed conformations for both α4β1 and α5β1 integrins are achieved by the ~25,000-fold slower off-rate of the EO conformation (*Figure 7C*). Similar to the differences in on-rates, the differences in off-rates can be understood in terms of the structural details in the ligand-binding pocket and the much higher affinity of the EO state. The tighter Asp sidechain binding pocket and greater burial of the Asp provide a barrier to dissociation (*Figure 7D*). The number of hydrogen bonds of the Asp sidechain to the β1-α1 loop backbone increases in the open state (*Zhu et al., 2013*). Furthermore, the greater burial of these polar bonds and increased network of hydrogen bonds around them increase their strength.

The intrinsic ligand-binding kinetics of integrin conformational states described here are consistent with previous kinetic observations. These studies showed that activating integrin ensembles with $Mn^{2+}$ or activating IgG or Fab, using conditions that in retrospect would partially but not completely shift integrin ensembles to the EO state, decreased the ligand off-rates of integrins α4β1 and α5β1 (*Chigaev et al., 2001*; *Takagi et al., 2003*). The extremely long lifetime of the α5β1 complex with fibronectin in the EO state, around several hours, also explains why the α5β1 complex with fibronectin in $Mn^{2+}$ was much more rapidly reversed by mAb 13 IgG specific for the closed conformations than by competitive inhibitor (*Mould et al., 2016*; *Mould et al., 2014*). One important difference from most previous work, including our own (*Takagi et al., 2003*), is our ability to fit all of our data to a 1:1 binding model. All our fits were global; that is, data at all ligand concentrations were used to create the fits in each panel shown in figures. This is the most rigorous way to fit data. When more variables are introduced such as a rate for conformational change to a different state or additional binding and dissociation rates, better fits can always be achieved. Notably, our data fit the 1:1 binding model whether one, two, or three states were present in the ensemble. Therefore, we believe that the rates of interconversion are relatively fast and attribute the difference from some previous results to homogenous protein preparations free from aggregates and homogenous incorporation of binding partners either as native proteins on intact cells or as purified proteins immobilized uniformly through single, site-specific biotins to streptavidin biosensors.

## Integrin activation

A major impetus for these studies was to determine the pathway for activation of integrins in cells, that is, the activation trajectory. Of key importance is how integrins on the cell surface first engage ligands. Integrin signaling is governed by cytoskeletal force and the force-stabilized, high-affinity, EO conformation is the only state competent to mediate cell adhesion (*Alon and Dustin, 2007*; *Astrof et al., 2006*; *Li and Springer, 2018*; *Li et al., 2017*; *Nordenfelt et al., 2016*; *Nordenfelt et al., 2017*; *Sun et al., 2019*; *Zhu et al., 2008*). Mechanotransduction occurs when an integrin binds to ligand anchored in the extracellular environment and at the same time its cytoplasmic domain binds to a cytoskeletal adaptor and links to actin retrograde flow, resulting in a tensile force transmitted through the integrin that stabilizes the EO conformation over the BC conformation. Thus, the on- and off-rates

of ligand binding to integrins are among the key parameters that determine the efficiency of cytoskeletal force regulation. We found that the closed states, with loose ligand-binding pockets, have higher on-rates for ligand binding, making them the most efficient state for encountering ligand.

What is the cell biological context in which we should think about our finding that the closed states are kinetically poised to bind ligand as the first extracellular event in integrin activation? And what is integrin inside-out signaling? This term was first used in a review on adhesion molecules of the immune system (*Springer, 1990*) after the discovery that integrins on lymphocytes (*Dustin and Springer, 1989*) and platelets (reviewed in *Hynes, 1992*) only became active in adhesion after these cells were stimulated through other receptors. The commonality is that lymphocytes and platelets in the bloodstream do not exhibit polarization or actin retrograde flow, but do so after stimulation, consistent with the model that integrins are activated by the tensile force transmitted through them between actin retrograde flow and extracellularly embedded ligands (*Li and Springer, 2017*) and findings that integrins are aligned with actin retrograde flow (*Nordenfelt et al., 2017*; *Swaminathan et al., 2017*) and transmit tensile force (*Nordenfelt et al., 2016*; *Sun et al., 2016*). Our results on the kinetics of ligand binding of different integrin conformational states are relevant to both integrins on cells in suspension as studied here and to unbound integrins on adherent cells since the kinetics are intrinsic to the EO, EC, and BC states.

Our results on ligand-binding kinetics provide a stepping stone to a remaining important question, the kinetics of integrin conformational change. Equilibria show that the ligand-bound open state is ~1000-fold more favored than the ligand-bound closed state. Thus, following step 1 of binding of ligand to the closed state, step 2 of conformational change to the more stable ligand-bound EO state should be rapid, but remains to be measured.

It should be emphasized that the populations of integrin conformational states we have measured on cells in suspension are for unliganded integrins. On adherent cells, integrins bound to ligand on the substrate belong to a separate pool of liganded integrins, which are removed from and not included in the populations of unliganded integrins we have measured. Ligand-bound integrins may dominate on the ventral surface of adherent cells, as emphasized by the predominance of extended integrin LFA-1 on the ventral surface of lymphoid cells bound to the LFA-1 ligand ICAM1, and the greater abundance of bent LFA-1 on the ventral surface of the same cells bound to the α4β1 ligand fibronectin (*Moore et al., 2018*).

Are the populations of unliganded integrin conformational states the same on suspended and adherent cells? We do not know the kinetics of binding of adaptors such as talin or kindlin to integrins or what fraction of unliganded integrins on suspended and adherent cells are bound to adaptors. The adaptor talin appears to interfere with association between integrin α and β subunit transmembrane/cytoplasmic domains in the BC state (*Lau et al., 2009*; *Zhu et al., 2009*), and thus if talin is bound to a higher proportion of unliganded integrins in adherent cells than suspended cells, the populations of unliganded EC and EO states might be increased relative to the BC state. Adaptor-bound unliganded integrins might also show preferential localizations in cells. Extended and unliganded β1 integrins have been shown to localize along the leading edge of fibroblast lamellae and growth cone filopodia to probe for adhesion sites (*Galbraith et al., 2007*). High-affinity, unliganded integrin αVβ3 was also shown to be recruited to lamellipodia in endothelial cell migration (*Kiosses et al., 2001*). Rapid ligand binding, together with rapid cytoskeletal adaptor binding, would enable their coincidence to regulate integrin activation, thus providing a seamless method for activating integrins at cellular locations where actin, talin, and kindlin are activated and extracellular ligand is available.

## Materials and methods

### Fabs

IgGs of 8E3 (*Mould et al., 2005*), 9EG7 (*Bazzoni et al., 1995*), 12G10 (*Mould et al., 1995*), HUTS4 (*Luque et al., 1996*), mAb13 (*Akiyama et al., 1989*), and SNAKA51 (*Clark et al., 2005*) were produced from hybridomas and purified by protein G affinity; Fabs were prepared with papain digestion in phosphate-buffered saline (PBS with 137 mM NaCl, 2.7 mM KCl, 10 mM Na₂HPO₄, and 1.8 mM KH₂PO₄, pH7.4) with 10 mM EDTA and 10 mM cysteine and papain: IgG mass ratio of 1:500 for 8 hr at 37°C, followed by Hi-Trap Q chromatography in Tris-HCl pH 9 with a gradient in the same buffer to 0.5 M NaCl. These conformation-specific Fabs are used to stabilize integrin α4β1 and α5β1 into

ensembles of defined conformational states. To ensure saturable population of target conformations, conformation-specific Fabs were used at concentrations (shown in the figure legends) well above the concentration giving half-maximum responses, that is, their EC50 values on each integrin preparation (Table S1 in *Li and Springer, 2018*; *Li et al., 2017*).

## Integrin α5β1 soluble preparations

Integrin α5β1 ectodomain (α5 F1 to Y954 and β1 Q1 to D708) with secretion peptide, purification tags, and C-terminal clasp (*Takagi et al., 2001*) were produced by co-transfecting the pcDNA3.1/Hygro(-) vector coding the α-subunit and pIRES vector coding the β-subunit into HEK 293S GnTI⁻/⁻ (*N*-acetylglucosaminyl transferase I-deficient) cells. Stable transfectants were selected with hygromycin (100 µg/ml) and G418 (1 mg/ml), and proteins were purified from culture supernatants by His tag affinity chromatography and Superdex S200 gel filtration after cleavage of C-terminal clasp and purification tags with TEV protease (*Li et al., 2017*).

## Peptidomimetic and macromolecule fragments

FITC-labeled α4β1-specific probe, 4-((N′-2-methylphenyl)ureido)-phenylacetyl-L-leucyl-L-aspartyl-L-valyl-L-prolyl-L-alanyl-L-alanyl-L-lysine (FITC-LDVP) and its unlabeled version (LDVP), were from Tocris Bioscience (Avonmouth, Bristol, UK). Human VCAM D1D2 (mature residues F1 to T202) were expressed and purified from HEK 293S GnTI⁻/⁻ cell line supernatants by affinity chromatography and gel filtration (*Yu et al., 2013*). VCAM D1D2 was fluorescently labeled with Alexa Fluor 488 NHS Ester (Thermo Fisher Scientific). Human Fn3$_{9\text{-}10}$ S1417C mutant (mature residues G1326 to T1509) and its synergy and RGD site (R1374A&P1376A&R1379A&S1417C&Δ1493-1496) mutated inactive version were expressed in *Escherichia coli* and purified as described (*Li et al., 2017*; *Takagi et al., 2001*). Fn3$_{9\text{-}10}$ S1417C mutant was fluorescently labeled with Alexa Fluor 488 C5 maleimide (Thermo Fisher Scientific) at residue Cys-1417. Both Fn3$_{9\text{-}10}$ S1417C mutant and its inactive version were biotinylated with Maleimide-PEG11-Biotin at residue 1417 (Thermo Fisher Scientific) in PBS.

## Cell lines

Jurkat and K562 cell lines were purchased from ATCC, which have been authenticated by STR profiling. Monthly mycoplasma contamination tests carried out in our lab confirmed the cells were free of mycoplasma.

## Quantitative fluorescent flow cytometry

Jurkat and K562 cells (10⁶ cells/mL in RPMI-1640 medium, 10% FBS) were washed twice with assay medium (Leibovitz's L-15 medium, 10 mg/mL BSA) containing 5 mM EDTA, twice with assay medium alone, and suspended in assay medium. Cells at 2 × 10⁶ cells/mL were incubated with indicated concentration of Fabs for 30 min at 22°C. Addition of FITC-LDVP, Alexa488-VCAM D1D2 (1.6 labeling ratio), or Alexa488-Fn3$_{9\text{-}10}$ (1.0 labeling ratio) at indicated concentrations initiated association. Association was measured as MFI at successive time points after addition of the fluorescent ligands. Addition of 500-fold higher concentration of the unlabeled ligand at the end of the association phase initiated the dissociation phase. Background MFI for FITC-LDVP, Alexa488-VCAM D1D2, and Alexa488-Fn3$_{9\text{-}10}$ in the presence of 10 mM EDTA was subtracted (*Figure 2—figure supplement 1*).

## Fitting flow cytometry and BLI kinetic binding traces with 1 vs. 1 Langmuir binding model

Kinetic traces including both the association phase and the dissociation phase at different analyte concentrations were globally fitted to the following function:

$$R_t = \left(\frac{1}{2} + \frac{(t_D\text{-}t)}{2|t_D\text{-}t|}\right) \frac{R_{\max} k_{on}[A]}{k_{off} + k_{on}[A]} \left(1\text{-}e^{-\left(k_{off} + k_{on}[A]\right)t}\right)$$
$$+ \left(\frac{1}{2} - \frac{(t_D\text{-}t)}{2|t_D\text{-}t|}\right) \frac{R_{\max} k_{on}[A]}{k_{off} + k_{on}[A]} \left(1\text{-}e^{-\left(k_{off} + k_{on}[A]\right)t_D}\right) e^{-k_{off}\left(t\text{-}t_D\right)},$$

where t is time, $R_t$ is response at time t, $t_D$ is the time that dissociation starts, [A] is the analyte concentration, and $R_{\max}$ is the maximum response. The first term fits the data in the association phase, and the second term fits the data in the dissociation phase. The prefactor of the first term is 1 prior to $t_D$ and becomes 0 after $t_D$, whereas the prefactor for the second term is 0 prior to $t_D$ and becomes 1 after

$t_D$. Nonlinear least-square fit of $R_t$, [A], and t to the above equation yields the on-rate, $k_{on}$, off-rate, $k_{off}$, and $R_{max}$.

## Biolayer interferometry

Binding kinetics of unclasped high-mannose α5β1 ectodomain and Fn3$_{9-10}$ was measured by BLI (*Wallner et al., 2013*) with streptavidin biosensors on an Octet RED384 System. The reaction was measured on 96-well plate (200 µL/well) in buffer with 20 mM Tris HCl (pH 7.4), 150 mM NaCl, 1 mM Ca$^{2+}$, 1 mM Mg$^{2+}$, and 0.02% Tween20. Streptavidin biosensors were hydrated in reaction buffer for 10 min before starting the measurements. Each biosensor was sequentially moved through five wells with different components: (1) buffer for 3 min in baseline equilibration step; (2) 35 nM biotin-Fn3$_{9-10}$ for 1 min for immobilization of ligand onto the biosensor; (3) indicated concentrations of Fabs for 5 min for another baseline equilibration; (4) indicated concentrations of α5β1 ectodomain and Fabs for the association phase measurement; and (5) indicated concentrations of Fabs for the dissociation phase measurement. Each biosensor has a corresponding reference sensor that went through the same five steps, except in step 2 the ligand was replaced with 35 nM inactive version of Fn3$_{9-10}$ with both the RGD-binding site and the synergy site (PHSRN) mutated. Background-subtracted response in both the association and dissociation phases, and at different α5β1 ectodomain concentrations, was globally fit to the 1 vs. 1 Langmuir binding model, with $k_{on}^{app}$ and $k_{off}^{app}$ as shared fitting parameters and maximum response ($R_{max}$) for each biosensor as individual fitting parameter. The equilibrium binding response ($R_{eq}$) was calculated from $k_{on}^{app}$ and $k_{off}^{app}$ values at each α5β1 ectodomain concentration and fit to a dose–response curve to calculate $K_d$ values as a check on $k_{off}^{app}/k_{on}^{app}$ values. To calculate $R_{eq}$, fitted $k_{on}$, $k_{off}$, and $R_{max}$ values at each α5β1 ectodomain concentration [A] were used to calculate $R_{eq}$ at a time 1000-folder longer than the 'binding time', that is, $t = 1000 * \frac{1}{k_{on}[A]}$ with the following equation:

$$R_{eq} = \frac{R_{max}k_{on}[A]}{k_{off}k_{on}[A]}(1 - e^{-(k_{off}+k_{on}[A])t})$$

## Calculating ligand-binding and dissociation rates for the BC and EC states

The measured on- and off-rates ($k_{on}^{app}$ and $k_{off}^{app}$) for each defined ensemble containing two or three states shown in *Figures 2–4* was approximated by the on- and off-rates of each state weighted by its population in the ensemble (*Figure 7A*, Eqs. 1–4). At steady state, the population of the free integrin states and the ligand-bound integrin states was calculated based on the previously determined population and intrinsic ligand-binding affinity of each state (*Figure 7—figure supplement 1*, Eqs. S5–S10) in the respective integrin α4β1 and α5β1 preparations (*Li and Springer, 2018*; *Li et al., 2017*; *Figure 7B*). Specifically, $\frac{K_a^{EO}}{K_a^{EC}}$ and $\frac{K_a^{EO}}{K_a^{BC}}$ (in *Figure 7—figure supplement 1B*, Eqs. S8–S10) are the intrinsic ligand-binding affinity ratios of the EO state and the closed states. For integrin α4β1, the ratios were averaged to 729 ± 211 from six α4β1 preparations, including α4β1 headpiece with high-mannose N-glycans, α4β1 ectodomain with high-mannose N-glycans, α4β1 ectodomain with complex N-glycans, and intact α4β1 on three different cell lines (*Li and Springer, 2018*); for integrin α5β1, the intrinsic ligand-binding affinity ratio of the EO state and the closed states were averaged to 3106 ± 1689 from eight soluble α5β1 preparations that varied in the presence or absence of the lower legs, of a loose clasp in place of the TM domain, and in whether the N-linked glycan was complex, high mannose, or shaved (*Li et al., 2017*). Using $k_{on}^{EO}$ and $k_{off}^{EO}$ rates experimentally measured in *Figures 2–4*, $k_{on}^{EC}$ and $k_{off}^{EC}$ were derived from the $k_{on}^{app(EC+EO)}$ and $k_{off}^{app(EC+EO)}$ measured in extended ensembles, respectively (*Figure 7A*, Eqs. 1 and 2). By including the values for $k_{on}^{EC}$ and $k_{off}^{EC}$ in addition to $k_{on}^{EO}$ and $k_{off}^{EO}$, $k_{on}^{BC}$ and $k_{off}^{BC}$ were then derived from $k_{on}^{app(BC+EC+EO)}$ and $k_{off}^{app(BC+EC+EO)}$ measured in basal ensembles, respectively (*Figure 7A*, Eqs. 3 and 4).

## Acknowledgements

We thank Kelly L Arnett at the Center for Macromolecular Interactions of Harvard Medical school for training and consultation on BLI measurement. We thank Taekjip Ha for suggestions on our manuscript. We thank Margaret Nielsen for making cartoons for each figure and for figure preparation. This work was funded by NIH R01 HL131729 ('Activation trajectories of integrin a5b1').

## Additional information

### Funding

| Funder | Grant reference number | Author |
|---|---|---|
| National Heart, Lung, and Blood Institute | R01-HL131729 | Jing Li<br>Timothy A Springer |

The funders had no role in study design, data collection and interpretation, or the decision to submit the work for publication.

### Author contributions

Jing Li, Conceptualization, Formal analysis, Investigation, Methodology, Writing - original draft, Writing – review and editing; Jiabin Yan, Formal analysis, Investigation; Timothy A Springer, Conceptualization, Funding acquisition, Supervision, Writing – review and editing

### Author ORCIDs

Jing Li  http://orcid.org/0000-0001-9603-7788
Timothy A Springer  http://orcid.org/0000-0001-6627-2904

### Decision letter and Author response

Decision letter https://doi.org/10.7554/eLife.73359.sa1
Author response https://doi.org/10.7554/eLife.73359.sa2

## Additional files

### Supplementary files

• Transparent reporting form

### Data availability

All data generated or analyzed during this study are included in the manuscript and source data files submitted.

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
