## [Editor Report]

This manuscript describes a detailed measurement and calculation of integrin ligand-binding kinetics, which are very important for the understanding of integrin activation. The data clearly indicate that low-affinity-binding states with closed conformation bind ligand with the mode of ‘fast on, fast off,’ while the high-affinity-binding state with the open conformation shows very slow off-rate. The kinetics measurements were well designed and a lot of work was done in this study.

---

## [Decision Letter]

**Decision letter after peer review:**

Thank you for submitting your article "Low affinity integrin states have faster binding kinetics than the high affinity state" for consideration by *eLife*. Your article has been reviewed by 3 peer reviewers, including Reinhard Fässler as the Reviewing Editor and Reviewer #1, and the evaluation has been overseen by José Faraldo-Gómez as the Senior Editor.

Essential revisions:

1) The ability of EO-binding Fabs to entirely shift the integrin ensemble to the EO conformation is a major assumption of the approach by Li, Yan and Springer. However, recently published structural data of native, full-length α5β1 integrin embedded in nanodiscs (Schuhmacher et al. (2021) Sci Adv) showed that not the entire integrin population assumes the EO conformation in the presence of FN37-10 and the EO-stabilizing TS2/16 Fab. This significant discrepancy requires clarification, without which the entire approach remains questionable.

2) The paper, in all its details, is quite difficult to understand for the broad audience of *eLife*, but also to the expert audience, as all calculations are based on the formula- and data-rich earlier publications. These previous papers contain e.g. a variety of different integrin constructs with different glycosylation (complex, high mannose, no glycosylation) and integrin ligands (cRGD, linear RGD, FN9-10). It was tedious to find the exact data that the authors refer to, and it is almost impossible to fully understand the current study without constantly looking up data from these past publications. Try to make it easier for the readers!

3) It appears that several experiments were only performed once. If this was the case, the authors must repeat measurements. Clearly mention, how often experiments were repeated.

---

## [Author Response]

Essential revisions:1) The ability of EO-binding Fabs to entirely shift the integrin ensemble to the EO conformation is a major assumption of the approach by Li, Yan and Springer. However, recently published structural data of native, full-length α5β1 integrin embedded in nanodiscs (Schuhmacher et al. (2021) Sci Adv) showed that not the entire integrin population assumes the EO conformation in the presence of FN37-10 and the EO-stabilizing TS2/16 Fab. This significant discrepancy requires clarification, without which the entire approach remains questionable.

There are two issues here. First, TS2/16 is not conformation specific. We published in Su et al., 2016 that TS2/16 Fab binds to all three states of a5b1. It appears to have higher affinity for the EO than closed states, but is clearly not conformation-specific, and we have therefore never used it in any of our studies on conformational ensembles, which exclusively use conformation state-specific Fabs. Our results on TS2/16 Fab were confirmed by the crystal structure of closed a6b1 bound to TS2/16 Fv-clasp and cryoEM of open a5b1 bound to TS2/16 Fv-clasp and Fn. The second issue is that Fn dissociates from the integrin because it is separated from both free Fn and TS2/16 Fv-clasp by gel filtration. The law of mass action requires free Fn and TS2/16 Fv-clasp to stabilize complex formation. The referee appears to refer to Figure 7 of Schumacher et al., which shows negative stain images. However, Fn3 7-10 is only visible in two of the three class averages of EO-like a5b1 in Figure 7C, lower right. And is probably not in any of the other images on the left in Figure 7C, which fail to show any evidence for Fn density.

In the Discussion, we now go into the issues of TS2/16, why we never used this antibody in any of our work on conformational ensembles, the law of mass action, and an erroneous claim that RGD cannot open the headpiece. I have been in touch with phone by Jun Takagi and Naoka Mizuno and explained to them that unfortunately it was necessary to explain these issues and they totally understood this. And Naoka also never claimed that Fn could bind to the closed conformation and said that it stabilized the open headpiece in her manuscript. Her manuscript goes into a discussion of the dynamics of integrin ligand binding and conformational change which is entirely consistent with our views. We are in good agreement on all the issues I discussed above.

2) The paper, in all its details, is quite difficult to understand for the broad audience of eLife, but also to the expert audience, as all calculations are based on the formula- and data-rich earlier publications. These previous papers contain e.g. a variety of different integrin constructs with different glycosylation (complex, high mannose, no glycosylation) and integrin ligands (cRGD, linear RGD, FN9-10). It was tedious to find the exact data that the authors refer to, and it is almost impossible to fully understand the current study without constantly looking up data from these past publications. Try to make it easier for the readers!

Thank you for pointing this out. When we cite previous publications, we now also include the figure panels where the data can be found. Also in Supplemental Figure 3 we show the populations of all conformational states in the many different types of integrin preparations we have studied. Both in absence of ligand and saturating concentrations of ligand. This should be a helpful look up table for other workers in the field.

3) It appears that several experiments were only performed once. If this was the case, the authors must repeat measurements. Clearly mention, how often experiments were repeated.

We clarified this in the figure legends. Each experiment was performed at least twice. The initial experiments were to test the range of variables that can lead to reliable fit of the data to get the kinetic parameters. These variables include analyte concentration used for the measurements in Figure 2, Figure 3 and Figure 4, Fab concentrations used for the measurements in Figures 5 and 6, and association phase length and dissociation phase length in Figure 2-6. The final data reported is the dataset acquired after optimizing all these variables and giving reliable fitting parameters through global fit of all the data points at different analyte concentrations in both association phase and dissociation phase.

Furthermore, each concentration is an independent experiment capable of determining the values we reported. By globally fitting to all the analyte concentrations used we obtained a single on and a single off rate, that reflect multiple independent determinations of an experiment shown. As we now emphasize, these multiple determinations that are globally fit with one another are also sensitive to any mistakes in the concentrations we used and the assumptions of our 1:1 binding model, since any discrepancies in these will increase the errors we determine. We show how our errors correspond to confidence intervals for the values we have determined. This is better than the typical statistics reviewers are used to, which show whether two groups are significantly different. Our confidence intervals show the confidence in the specific k-off and k-on values we have determined.